# Genomics of Dwarfism in Italian Local Chicken Breeds

**DOI:** 10.3390/genes14030633

**Published:** 2023-03-03

**Authors:** Francesco Perini, Filippo Cendron, Zhou Wu, Natalia Sevane, Zhiqiang Li, Chunhua Huang, Jacqueline Smith, Emiliano Lasagna, Martino Cassandro, Mauro Penasa

**Affiliations:** 1Department of Agricultural, Food and Environmental Sciences, University of Perugia, 06121 Perugia, Italy; 2Department of Agronomy, Food, Natural Resources, Animals and Environment, University of Padova, 35020 Legnaro, Italy; 3The Roslin Institute and Royal (Dick) School of Veterinary Studies, University of Edinburgh, Easter Bush, Midlothian EH25 9RG, UK; 4Department of Animal Production, Veterinary Faculty, Universidad Complutense de Madrid, Avenida Puerta de Hierro, 28040 Madrid, Spain; 5College of Animal Science and Technology, Chengdu Campus, Sichuan Agricultural University, Chengdu 611130, China; 6Federazione delle Associazioni Nazionali di Razza e Specie, 00187 Roma, Italy

**Keywords:** genomic region, indigenous breed, chicken, SNP, GWAS

## Abstract

The identification of the dwarf phenotype in chicken is based on body weight, height, and shank length, leaving the differentiation between dwarf and small breeds ambiguous. The aims of the present study were to characterize the sequence variations associated with the dwarf phenotype in three Italian chicken breeds and to investigate the genes associated with their phenotype. Five hundred and forty-one chickens from 23 local breeds (from 20 to 24 animals per breed) were sampled. All animals were genotyped with the 600 K chicken SNP array. Three breeds were described as “dwarf”, namely, Mericanel della Brianza (MERI), Mugellese (MUG), and Pepoi (PPP). We compared MERI, MUG, and PPP with the four heaviest breeds in the dataset by performing genome-wide association studies. Results showed significant SNPs associated with dwarfism in the MERI and MUG breeds, which shared a candidate genomic region on chromosome 1. Due to this similarity, MERI and MUG were analyzed together as a meta-population, observing significant SNPs in the *LEMD3* and *HMGA2* genes, which were previously reported as being responsible for dwarfism in different species. In conclusion, MERI and MUG breeds seem to share a genetic basis of dwarfism, which differentiates them from the small PPP breed.

## 1. Introduction

With the development of poultry genetics and breeding, selection and crossbreeding have shaped the phenotypic diversity and demographic history of local breeds [1,2]. Moreover, their domestication has strongly limited the phenotypic differentiation of local breeds by promoting genetic variants favorable to productive traits [3]. Among them, animal body size is the paramount trait determining the profitability of poultry meat; indeed, optimizing this trait has been a major goal during domestication [3,4,5]. Hence, a heavy body weight and rapid growth have always been important traits within the poultry industry. In response to natural and/or artificial selection, chicken breeds vary enormously in size. In contrast, in response to natural and/or artificial selection, chicken breeds vary enormously in size, also allowing the development of much lighter chicken breeds called dwarf breeds. Dwarf animals are characterized by short body stature and light body weight [6]. However, the identification of dwarf animals based on phenotypic evaluation is not easy, as several “non-heavy” breeds only have small stature and are not actual dwarfs [6]. In chickens, the dwarf phenotype is evaluated by different traits including, but not limited to, body weight, height, and shank length [6,7]. Based on physiological and genetic characteristics, four distinct types of dwarfism in chickens have been identified: sex-linked dwarfism, which has been well-studied and is caused by mutations in the growth hormone receptor gene (*GHR*) [8,9]; autosomal dwarfism, which is caused by a nonsense mutation in the transmembrane protein 263 gene (*TMEM263*) [10]; the bantam phenotype, associated with the *HMGA2* gene on chromosome 1 [11]; and creeper dwarfism due to a creeper allele described as a 25 kb deletion containing the *IHH* and non-homologous end-joining factor 1 (*NHEJ1*) genes [12,13].

In Italy, the indigenous breed Mericanel della Brianza (MERI), from the Lombardia region, is the most popular dwarf breed and derives from dwarf chickens that were spread across small rural farms at the beginning of the last century [14]. The MERI is the only Italian bantam chicken breed with an official standard recognized by the Italian Association of Fancy Breeders and is mainly reared on fancy farms in the Lombardia region [15]. Indeed, the breed standard is based on qualitative traits such as comb type and plumage, shank, foot, skin, and eye color. Other Italian local breeds such as the Pepoi (PPP) and the Mugellese (MUG) are presumed to be dwarf according to their body size and the other quantitative traits reported in Table 1.

The PPP breed is mainly spread across the Veneto region and is reared for fancy but also for productive purposes. In fact, the PPP has good breast muscle mass, providing very tasty meat (https://www.pollitaliani.it/razze/pepoi/, accessed on 21 December 2022). The MUG chicken is a small breed that originated from the Mugello area in the northeast of the Tuscany region. The MUG is widespread in the region due to its excellent aptitude for brooding; indeed, it has been traditionally used as a putative hen for eggs of other chicken breeds. Currently, the MUG is under a recovery selection program due to its small effective population size [16,17].

The MERI, MUG, and PPP are similar in their genetics and background [18]; however, the body weight and morphological traits are more similar between MERI and MUG [15,19]. To the best of our knowledge, the genomic features linked to dwarfism of the three breeds have not been investigated thus far. The aims of the present study were to characterize the genome-wide sequence variation associated with the dwarf phenotype in MERI, MUG, and PPP, and to investigate the genes associated with their phenotype. Understanding the genetic differences underlying the dwarf phenotype is important to assess a marker-assisted genomic selection to maintain this phenotype.

## 2. Materials and Methods

### 2.1. Ethical Statement

Ethical approval was not required for the current study. Blood samples were collected in compliance with the European rules [Council Regulation (EC) No. 1/2005 and Council Regulation (EC) No. 1099/2009] during routine health controls by the public veterinary service.

### 2.2. Samples Collection

The dataset was previously described in Cendron et al. [18] and consisted of 582 samples: 20 to 24 animals (equal number of males and females) for each of the 23 local breeds, and nine to 13 animals for each of the four commercial lines. From the total dataset, we extracted three breeds (MERI, MUG, and PPP; Table 1) supposed to be dwarf, and hence used as the case groups. Moreover, a control population was chosen that included the three heaviest local breeds, namely, Ermellinata di Rovigo (PER), Robusta Lionata (PRL), Robusta Maculata (PRM) (Table 1), and the Broiler Ross 708 (Broiler, 13 animals). Blood samples (2 mL) were collected from the ulnar vein and stored in Vacutainers^®^ tubes containing EDTA as an anticoagulant. All studied breeds were collected from at least three different poultry farms and conservation centers to ensure a representative sampling of the breed.

### 2.3. Genotyping

DNA extraction and genotyping were performed at Neogen (Ayr, Scotland, UK) using a commercial kit and the Affymetrix Axiom 600 K Chicken Genotyping Array, representing 580,961 SNPs. The Gallus_gallus 5.0 chicken assembly (accession number: GCA_000002315.3) was used in this study as the reference genome [20] including the markers located on chromosomes 1 to 28 and on the sex chromosomes. SNPs with a call rate < 95%, minor allele frequency < 5% and animals with more than 10% of the missing genotypes were removed from the dataset. The software PLINK 1.9 [21] was used to process the data.

### 2.4. Population Structure Analysis

The population structure of the final dataset was estimated with principal component analysis (PCA) and multidimensional scaling plot (MDS). First, we used the PLINK command “--keep” to extract the MERI, MUG, and PPP breeds and the control population (PER, PRL, PRM, and Broiler) from the .ped file. This subset was used for the population structure analysis performed through PCA, MDS, and Neighbor-Joining Tree (NJ). PLINK 1.9 was used for this purpose and to generate eigenvectors and eigenvalues, and the ggplot2 (v3.1.0) package of R software was used to visualize the results of PCA [22]. The 1-ibs distance matrix was computed in PLINK 1.9, converted to a NJ tree using the VCF2Dis software (https://github.com/BGI-shenzhen/VCF2Dis, accessed on 1 November 2022), and visualized with R package “treeio” [23]. Pair-wise genetic relationships within and between breeds were estimated using a matrix of genome-wide identity-by-state genetic distances in PLINK 1.9 and plotted using MDS (components C1 and C2).

### 2.5. GWAS Analyses

A genome-wide association study (GWAS) was performed, comparing the case (supposed dwarf breeds) and control populations. The MERI, MUG, and PPP breeds do not have normal-size chicken counterparts. The breeds used in the control population were chosen based on phenotypic data such as live body weight, wingspan, body length, breast width, shank width, and shank length [19]. Particularly, the live body weight was used as a discriminant for choosing the control population. Breeds with higher mean live body weight were chosen (Table 1) [19]. Moreover, the genetic features were evaluated according to the PCA, MDS, and NJ results, and the most divergent genetic backgrounds from the case population were categorized as control population. All breeds in the present study had no genetic outliers and showed clear clustering.

We performed GWAS using a single dwarf breed as well as all of the dwarf breeds combined, compared against the control population. GWAS was performed in GEMMA software v0.94.128 with linear-mixed models, adjusting for the effect of sex and relatedness, using a previously calculated matrix [24]. Upon the separate analysis of males and females, we observed no difference in male/female SNPs identified by the GWAS approach. Thus, further analyses combined males and females together. The genome-wide significance threshold was determined by the Bonferroni method, in which the conventional *p*-value (0.05) was divided by the number of tests performed as follows: cutoff = −log_10_(0.05/number of variants). A genomic control inflation factor, the lambda was calculated in each study to evaluate the confounding due to population stratification. Variants were then annotated using snpEFF version 5.1 with default parameters in vCard File (VCF) [25]. Only SNPs with significant *p*-values were taken into consideration for further analyses. The VCF file from the previous step was then used as input in Ensembl Variant Effect Predictor (VEP) to evaluate the effect of SNPs on the genes, transcripts, and protein sequence. *F*st values for single SNP loci observed between the experimental and control populations were calculated with PLINK 1.9. In order to compare the linkage disequilibrium (LD) around the *HMGA2* gene, we computed the r2 between the markers of this region against the highest associated variant. Moreover, Haploview software [26] was used for computing haplotypes around the *HMGA2* gene in the studied populations. The SNPs used for this purpose had MAF >0.10 and the Haploview software calculated the block (alleles) if 95% of informative (i.e., non-inconclusive) comparisons were in strong LD.

## 3. Results

### 3.1. Population Structure

In Figure 1a and in Appendix A, the MUG and MERI birds were well-distinguished yet very close to each other. On the other hand, the PPP birds—the third breed categorized as dwarf—were distant to the other two dwarf breeds in both PCA and MDS (Figure 1a and Appendix A). Additionally, according to the phylogenetic analysis, the relative genetic relationships within the breeds were estimated, which showed a similar separation of PCA and MDS results in both the dwarf breeds and control population, respectively. Figures representing the PCA and MDS results showed a similar distribution of the breeds along the graph, thus testifying to the capacity of the two methods to distinguish the breeds on a genomic basis. The clear separation among the breeds was confirmed in Figure 1b, where the three “dwarf” breeds shared a common ancestor before the PPP birds diverged from the other two breeds.

### 3.2. GWAS

Based on the genetic information, we performed independent GWAS for each of the three dwarf breeds using genetic variants against the control population (Figure 2). The values of lambda for all GWAS analyses were close to one (1.074 to 1.146), suggesting a good control in the population stratification. With regard to sex-linked dwarfism, which is caused by mutations in the *GHR* gene located on chromosome Z, no significant SNPs in or around this gene were detected (Appendix A). Notably different signals were observed among the three “dwarf” breeds (Figure 2a–c). With regard to the comparison of the MERI breed with control population, 643 SNPs passed the significance threshold (5 × 10^−8^), of which 215 were associated with known genes (Figure 2a). For the MUG breed, 256 variants with significant *p*-values and 96 genes associated with them were observed (Figure 2b). Finally, in Figure 2c, a large number of variants were significant (1379) for the PPP breed. Some of the genes detected in the MERI and MUG birds are known to be relevant to the dwarfism phenotype. Particularly in MUG birds, two main peaks on GGA1 and GGA5 were appreciable. For the peak on chromosome 5, the genetic variants were located in the intronic region of the *RGS6* gene. This was highlighted in all three supposed dwarf breeds.

### 3.3. Chromosome 1 Candidate Region

First, we focused on the SNPs on chromosome 1 in each of the three breeds. Although the MUG and MERI chickens did not share the same variants, they both showed the most significant SNPs in regions of chromosome 1 containing genes of interest for dwarfism. These include *HMGA*, *GRIP1*, and *LEMD3* in the MUG birds, and *WIF1*, *RASSF3*, and *SRGAP1* in MERI (Appendix A). For instance, the AX-75432737 SNP in MUG has been found at position 1:34378326 and it is responsible for an intronic variant within the *HMGA2* gene (*p*-value < 4.67^−18^; Appendix A). This region (1:33701467–34603740) did not seem to be relevant in the PPP breed. For this reason, we carried out a meta-analysis combining both MUG and MERI as a unique dwarf “population” and compared them against the control population (Figure 3a). Meta-analysis showed 99 significant variants (*p*-value < 5 × 10^−8^) annotated in 43 genes (Table 2).

Genes of interest such as *LEMD3* and *HMGA2* were also observed in the meta-analysis. Figure 3b clearly highlights the relevance of GGA1 in the meta-population Mericanel della Brianza + Mugellese (MM) analysis, with significant *p*-values for the variants associated with the *HMGA* and *LEMD3* genes. Clear implication of these two genes in dwarfism has been widely reported in the literature. In the same above-mentioned genomic region on GGA1, it was possible to find numerous genes linked with dwarfism, which were also observed in the MUG and/or MERI breeds [11,27,70]. For example, the AX-75431832 and AX-75432337 variants were commonly shared in MUG and MERI, and were mapped to intergenic regions between *TBC1D30-WIF1* and *MSRB3-HMGA2*, respectively. Moreover, *GRIP1* showed significance in the same genomic region in MUG and *RASSF3* in the MERI birds. Hence, in both the single breed analysis and meta-analysis, we observed a potential candidate region on chromosome 1. To explore population differentiation and genetic distance based on the GWAS results, we calculated the *F*st for each SNP in the dwarf breeds against the control population. We then took the significant SNPs associated with dwarf genes and filtered on the *F*st results (Appendix A). It was encouraging to note that the *F*st values of these SNPs were very close to 1, which indicated that these SNP loci and their associated dwarf genes had a great degree of differentiation in comparison to the control population. The LD data provided linkage information regarding the GGA1 locus. Appendix A reports the *p*-values from the GWAS and the r2 values for each SNP, in both the MERI and MUG analyses as well as the MM meta-analysis. Moreover, an investigation of haplotypes in the MUG and MERI populations was carried out by Haploview around *HMGA2* (1:34284446–34472431). Only MUG showed a unique block in *HMGA2* (Appendix A) and the frequency of this haplotype was 0.437 (Appendix A). MERI birds showed four blocks (Appendix A), of which the fourth was the longest haplotype and had a high frequency in population (i.e., 0.521) (Appendix A).

### 3.4. Other Candidate Regions

The other interesting peak seen in Figure 3a was located on GGA5, with variants located around the *RGS6* gene. Four intronic variants mapped to *RGS6* exhibited *p*-values that ranged from 2.7 × 10^−16^ to 6.41 × 10^−12^ (Appendix A). Of considerable importance is also the *IGF1R* gene, which was detected in the MUG analysis. *IGF1R* is located on chromosome 10 and is notoriously associated with reduced growth and dwarfism. *IGF1R* was observed thanks to a significant intronic variant in MUG birds, and it was also corroborated by the MM meta-analysis where upstream gene variation was noted. In both cases, suggestively significant *p*-values were observed, relative to the previously mentioned variants on chromosome 1. It was interesting to see the genes that were shared between the different analyses in this study. The Venn diagram in Figure 4a shows the candidate genes shared between the single breeds and meta-analysis, while Figure 4b depicts the same data without the genes annotated for the PPP analysis, and all the genes from Figure 4 are listed in Appendix A. Interestingly, only the calcitonin receptor (*CALCR*) gene was commonly shared among all groups (Figure 4a). In Figure 4b, however, eight genes were in common between the MERI, MUG and MM analyses. Among them, the gene currently known to be most related to the dwarf phenotype was *HMGA2*, followed by *USH2A*, *USH1C*, *RGS6*, *PDE8A*, *RRN3*, and *STXBP4*. Unfortunately, none of these genes are known to be related to dwarfism, although their function in relation to this phenotype has not been fully explored thus far.

## 4. Discussion

Chicken dwarf breeds were established in Europe from Greek and Roman times, only being used as fancy, luxury birds [71]. However, over time, there has been enormous development of these birds to act as brooder hens for similar birds such as pheasants and francolins. Unfortunately, for Italian dwarf chicken breeds, there are no documents ascertaining their history, origin, and which breeds were used for their establishment. Indeed, it is still difficult to categorize them as dwarf breeds. Only MERI is officially recognized as a dwarf breed, however, the genomic evidence for the dwarf phenotype is unclear. The MUG breed is recognized as an official breed with a standard phenotype; indeed, its morphometric measurements are similar to those of MERI birds. In contrast, PPP is not recognized either as an official breed or as a dwarf one. Additionally, the morphometric measurements of PPP are the most divergent among the three breeds in this study. To our knowledge, none of these three breeds has been studied for its genomic characteristics, apart from run-of-homozygosity and phylogenetic studies [18,72]. Even if the breeds taken into account have different backgrounds, the methods used for genomic distinction showed the ability to differentiate the breeds. Tight clustering reflects the genomic relationship within each breed, with no samples found as outliers.

### 4.1. Candidate Genes from GWAS

Single breeds showed similar signals on GGA5 compared to the control population. All GWAS analyses shared a common annotated gene (*RGS6*) in this region (5:26600000–27300000). The *RGS6* gene negatively regulates G protein signaling by activating GTPase activity. Although the precise physiological roles of *RGS6* are unknown, this gene is linked to cellular stress responses and induces cell cycle arrest and apoptosis [73]. In the same genomic region, another two genes were detected: *DPF3* in MM, MERI, and PPP, and *SIPA1L1* in MM and MERI. *SIPA1L1* is the signal induced proliferation associated 1-like 1 gene that has been implicated in osteogenesis in vitro [50] and has a strong association with maximal clutch length in chicken [74]. The most interesting gene related to the development of tissues in this region is double PHD fingers 3 (*DPF3*), which has a role in heart development. The *DPF3* gene is specifically expressed during heart development in mice, chickens, and zebrafish, and the knockdown of *DPF3* in zebrafish resulted in myocardial contractility and incomplete heart formation; thus, *DPF3* was categorized as a major gene for heart development, but it is also involved in skeletal muscle [68,75]. Another important gene related to dwarfism and growth is *IGF1R*, which was annotated with upstream genetic variation in the MM and MUG analyses. Reduced *IGF1R* expression has been reported in dwarf Holstein calves [6] and also in chicken [56].

The most important findings in the present study come from the identified region on chromosome 1. The *HMGA2* gene is significant in MERI, MUG and the MM meta-analysis, but not in PPP. This gene has a role in dwarfism in different species such as chicken [11], rabbit [70], and in humans [76]. Moreover, Plassais et al. [77] reported that *HMGA2* is the main candidate gene for body size in dogs. *HMGA2* controls the proliferation of myoblasts and skeletal muscle development by modulating the expression of IGF2-binding protein 2 (*IGF2BP2*), which regulates the downstream signaling of many genes related to cell growth [78]. In this study, none of the variants associated with the *HMGA2* gene were responsible for modification in the translated protein. Indeed, they were all intronic or intergenic variants. In another study in humans, the role of *LEMD3* in the pathogenesis of osteopoikilosis and short stature was highlighted [79]. *LEMD3* is located in the same genomic region as *HMGA2* and was identified in MM and MUG analyses in the present study. Carneiro et al. [70] found these genes to be associated with dwarfism in rabbit. Moreover, the *LEMD3* functional mechanism is an antagonist to transforming growth factor-β signaling (TGF-β) [80]. *LEMD3* has also been reported in the modulation of ear size in pigs [81], underlying the principal role in the modulation of growth in different species. Together with *LEMD3*, the *WIF1* gene has been reported in pigs, rabbits, and dogs as a candidate gene for the modulation of growth [27]. *WIF1* binds to Wnt proteins and inhibits the activity of the Wnt/b-catenin pathway. The Wnt/b-catenin pathway may regulate proliferation and differentiation in many tissues including controlling the growth of connective tissue by regulating connective tissue growth factor (CTGF) and fibroblast cells in the skin [70,77,81]. Regarding the present study, the *WIF1* gene was reported with an intronic variant at position 1:34008227 in the MM and MERI analyses.

### 4.2. Genomic Convergence on Chromosome 1

Our results highlight that the Fst values of the variants on GGA1 segregate in the control population. The Fst analysis evaluated the important differences that exist between the populations for the studied genomic region on GGA1. A different view of what happens on chromosome 1 was provided by the LD results. Appendix A depicts the LD levels of the most significant SNP compared with others from the same genomic region. High LD levels were found in the MERI genome, especially when compared to MUG and the MM meta-analysis. Moreover, the MUG birds showed distinct haplotypes around the *HMGA2* gene, suggesting a strong selection in that part of the genome. These results were in line with those reported by the Italian Association of Fancy Breeders for these breeds (https://www.fiavinfo.eu/nana-italiana-perche-no/ accessed on 10 January 2023). In fact, the MERI is the only breed to be officially recognized both as an official breed and at the same time as a dwarf. This result underlines that, over time, crossbreeding has led to selective pressure in MERI birds, precisely in the identified genes on chromosome 1. Regarding the PPP birds, these candidate genes were not identified in our analyses. In fact, it is reported that PPP is not an officially recognized breed because of its diverse phenotypic variability. Finally, the MUG breed seems to have different variants with respect to breeds with normal size. Based on the LD results, however, the chromosome 1 locus does not appear to be as significant as it seems to be for the MERI breed. Generally, the SNP arrays were designed with variants mostly mapped to non-coding genomic regions. Indeed, in the 600 K Affymetrix^®^ Axiom^®^ array for chickens, only 3.7% of SNPs out of 580,954, were in the coding regions [82]. This could explain why we have not found a causative mutation in an exonic region (e.g., in *HMGA2*), which may lead to a change in the final protein. However, this work has highlighted candidate genes for the dwarfism phenotype in MUG and MERI chickens. Further studies with deeper or targeted sequencing of the candidate regions could shed light on the main features of genes in the studied breeds.

## 5. Conclusions

In conclusion, we identified candidate genes for dwarfism in MUG and MERI chickens and highlighted the importance of genes on chromosomes 1 and 5, while we showed that the PPP population does not appear to be a true dwarf breed when we consider the candidate genomic loci and morphological data. Further studies should focus on the validation of results obtained in the present manuscript through a larger sample size and the genotyping of identified candidate genes.

## Figures and Tables

**Figure 1 genes-14-00633-f001:**
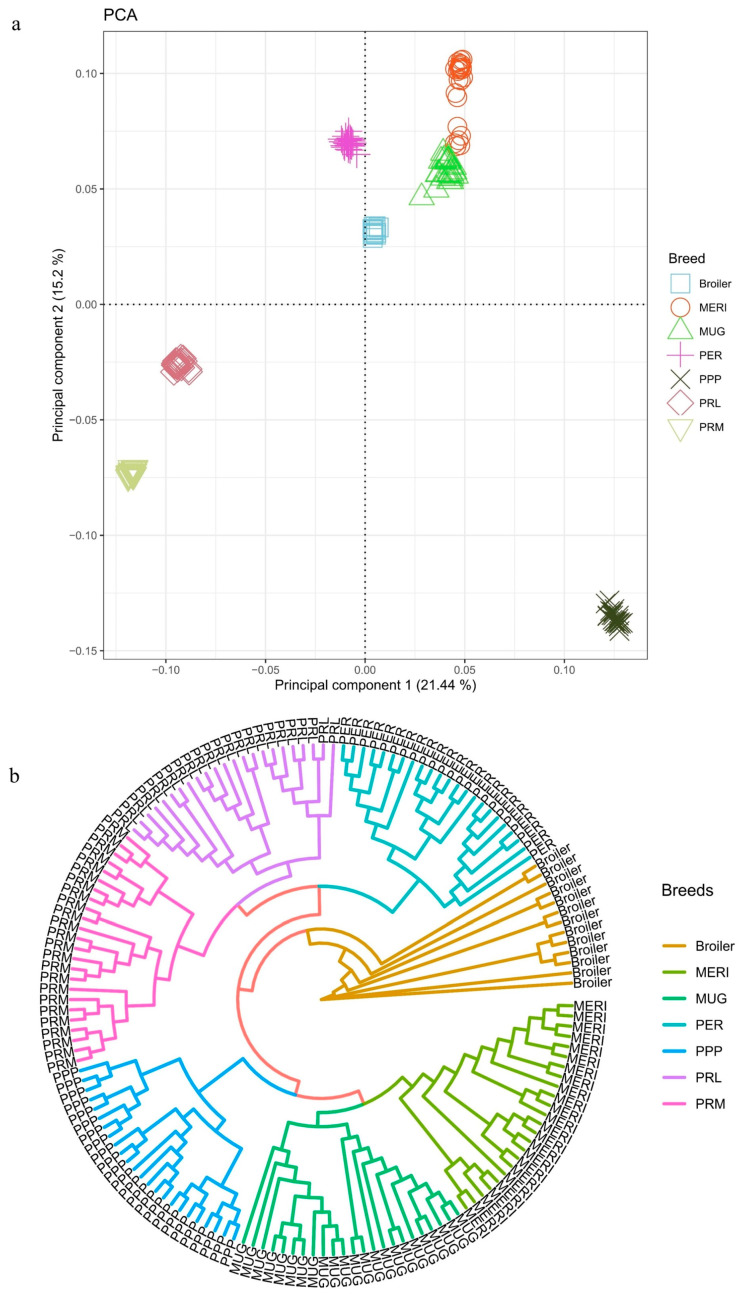
(**a**) Principal component analysis (PCA) and (**b**) Neighbor-Joining Tree (NJ) of the dwarf breeds and control population. Different breeds are highlighted with different shapes and colors: Mericanel della Brianza (MERI), Mugellese (MUG), Ermellinata di Rovigo (PER), Pepoi (PPP), Robusta Lionata (PRL), Robusta Maculata (PRM), and Broiler Ross 708 (Broiler).

**Figure 2 genes-14-00633-f002:**
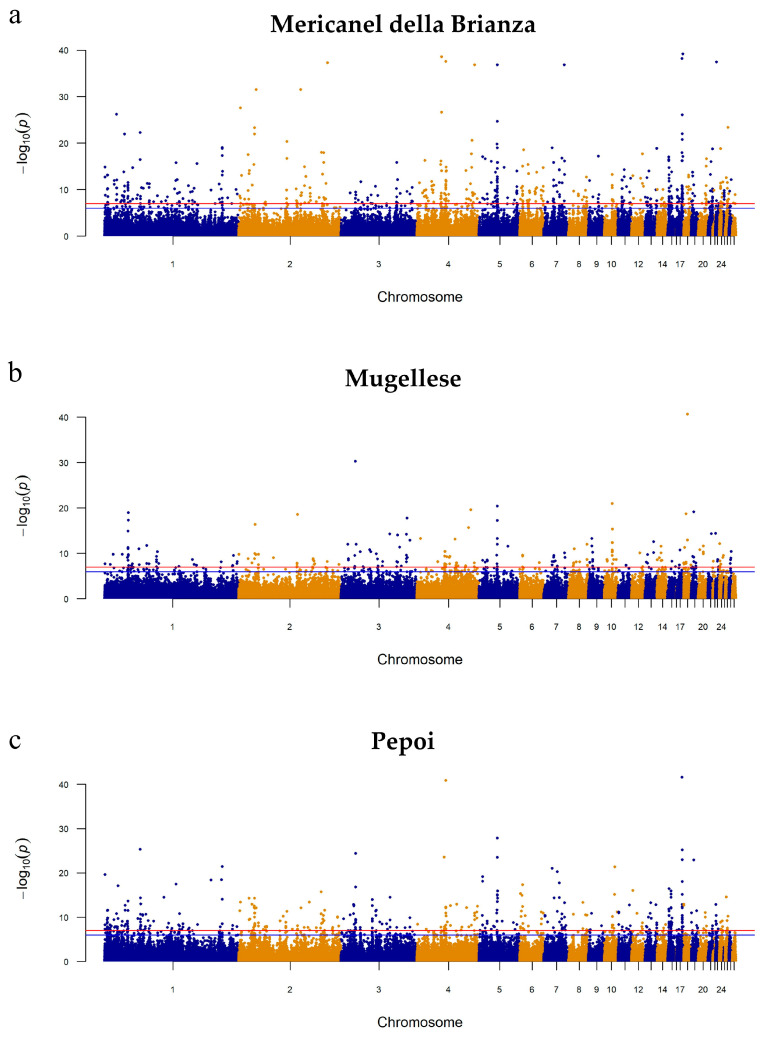
Manhattan plot of (**a**) Mericanel della Brianza, (**b**) Mugellese, and (**c**) Pepoi against the control population. The red line represents the *p*-value < −log10(1 × 10^−7^). The blue line represents suggestive significance, with a value of −log10(1 × 10^−6^).

**Figure 3 genes-14-00633-f003:**
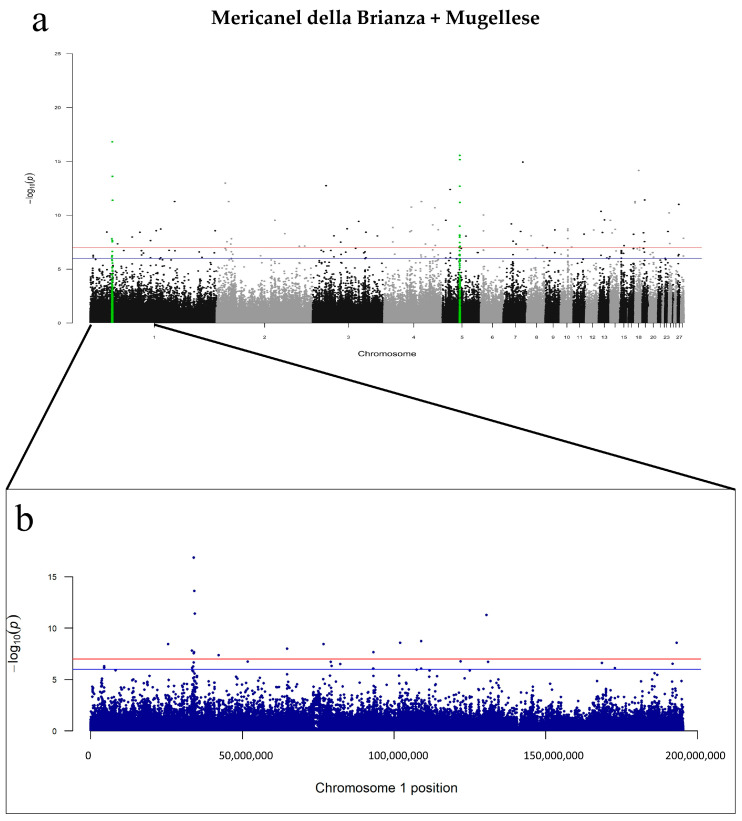
Manhattan plot of (**a**) Mericanel della Brianza + Mugellese breeds tested against the control population and (**b**) details of the chromosome 1 region. The red line represents the *p*-value < −log10(1 × 10^−7^). The blue line represents the suggestive significance, with a value of −log10(1 × 10^−6^).

**Figure 4 genes-14-00633-f004:**
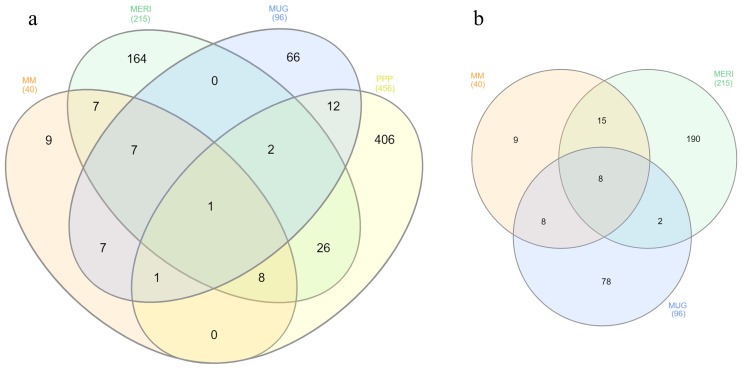
Venn diagrams of the annotated genes with significant variants from the GWAS analyses. (**a**) Venn diagram from the meta-analysis (MM), Mericanel della Brianza (MERI), Mugellese (MUG), and Pepoi (PPP) breeds and the (**b**) Venn diagram representing genes from the MM, MERI, and MUG breeds.

**Table 1 genes-14-00633-t001:** Mean and standard deviation (SD) of the morphometric traits of Mericanel della Brianza, Mugellese, and Pepoi local chicken breeds (presumed dwarf), and the local heavy breeds used as the control population, namely, Ermellinata di Rovigo, Robusta Lionata, and Robusta Maculata (Broiler Ross 708 is not reported in the table as phenotypic characterization was conducted only for local breeds).

Breed	Sex	Body Weight (g)		Body Length (cm)		Shank Length (cm)		Shank Width (cm)		Breast Width (cm)		Wingspan (cm)	
		Mean	SD	Mean	SD	Mean	SD	Mean	SD	Mean	SD	Mean	SD
Mericanel della Brianza													
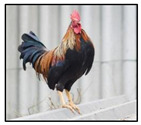	♂ (n = 6)	962.1	175.3	33.0	2.9	4.9	0.6	3.8	0.6	26.0	3.4	28.8	2.4
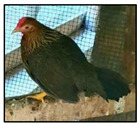	♀ (n = 17)	731.7	91.3	28.1	1.9	4.8	0.4	3.1	0.2	24.4	2.3	25.5	1.6
Mugellese													
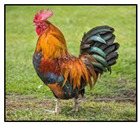	♂ (n = 7)	1082.1	91.5	29.0	1.6	6.2	0.3	3.9	0.3	26.1	1.0	32.0	2.0
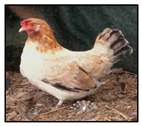	♀ (n = 17)	717.4	100.8	24.9	1.7	5.0	0.5	3.3	0.3	22.5	1.3	27.2	2.3
Pepoi													
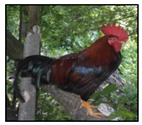	♂ (n = 12)	1860.0	182.1	36.9	1.3	9.4	0.5	4.6	0.4	33.4	3.2	43.0	1.8
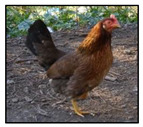	♀ (n = 12)	1293.3	219.2	32.3	1.6	7.5	0.4	3.8	0.3	28.9	1.9	37.2	1.8
Ermellinata di Rovigo													
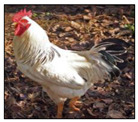	♂ (n = 12)	3436.7	216.2	45.7	1.4	11.0	0.3	5.9	0.1	39.1	1.3	53.5	0.6
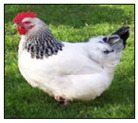	♀ (n = 12)	2322.5	152.5	40.0	1.9	9.1	0.6	4.8	0.2	34.8	1.6	45.9	1.3
Robusta Lionata													
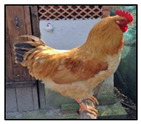	♂ (n = 12)	4035.8	1024.6	46.8	1.6	11.3	1.0	6.1	0.2	41.2	3.9	53.4	1.6
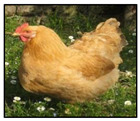	♀ (n = 12)	2753.3	378.0	41.4	2.4	9.3	0.7	4.7	0.2	36.9	2.9	46.7	1.7
Robusta Maculata													
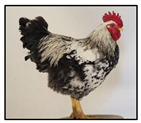	♂ (n = 12)	4221.7	253.2	38.7	2.2	9.5	0.7	4.7	0.2	37.3	3.5	46.2	1.8
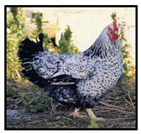	♀ (n = 12)	2831.7	450.6	45.6	2.1	11.5	0.8	6.0	0.3	42.3	3.6	54.1	1.4

**Table 2 genes-14-00633-t002:** Gene ID, name, function, and reference for genes annotated in the meta-analysis (MM).

Gene ID	Gene name	Function	Reference
*LEMD3*	*LEM domain containing 3*	Responsible for bone density disorders	[27]
*HMGA2*	*high mobility group AT-hook 2*	Contributes to vascular development and sprouting angiogenesis by promoting IGFBP2 production	[28]
*NTS*	*neurotensin*	Induces hypotension, hyperglycemia, and cyanosis	[29]
*MAEL*	*maelstrom spermatogenic transposon silencer*	Involved in the development and progression of bladder, liver, and colorectal cancers	[30]
*ILDR2*	*immunoglobulin-like domain containing receptor 2*	Negative regulator for T cells	[31]
*ERG*	*ETS transcription factor ERG*	Member of the E-26 transformation-specific (ETS) family of transcription factors with roles in development that include vasculogenesis, angiogenesis, hematopoiesis, and bone development	[32]
*MALRD1*	*MAM and LDL receptor class A domain containing 1*	Positively affects FGF15/19 levels, a hormone that can modulate bile acid levels, repress gluconeogenesis and lipogenesis, and promote glycogen synthesis	[33]
*CALCR*	*calcitonin receptor*	Involved in regulating follicular maturation in the chicken ovary	[34]
*USH2A*	*usherin*	Can cause Usher syndrome type 2 and non-syndromic retinitis pigmentosa	[35]
*PDE10A*	*phosphodiesterase 10A*	Its repression brings a reduction in muscle pathology and improvement in locomotion, muscle, and vascular function	[36]
*AMMECR1*	*AMMECR nuclear protein 1*	Its inactivation is associated with growth, bone, and heart alterations	[37]
*ENPEP*	*glutamyl aminopeptidase*	Encodes glutamyl aminopeptidase, which is related to tumorigenesis and immune microenvironment	[38]
*CASP6*	*caspase 6*	Expressed in gastric and colorectal cancers	[39]
*SGCZ*	*sarcoglycan zeta*	A factor in the pathogenesis of muscular dystrophy and is expressed mainly in vascular smooth muscle	[40]
*STX18*	*syntaxin 18*	Physically interacts with proteins involved in cell cycle and apoptosis	[41]
*MSX1*	*msh homeobox 1*	Its interaction with p53 inhibits tumor growth by inducing apoptosis and inhibits angiogenesis	[42]
*JAKMIP1*	*janus kinase and microtubule interacting protein 1*	It is a microtubule-associated protein predominantly expressed in neurons and lymphoid cells, which contributes to the establishment of neuronal morphology	[43]
*TACC3*	*transforming acidic coiled-coil containing protein 3*	Functions in mitotic spindle assembly and chromosome segregation	[44]
*ELP4*	*elongator acetyltransferase complex subunit 4*	Associated with language impairment, autism spectrum disorder, and mental retardation	[45]
*USH1C*	*USH1 protein network component harmonin*	Mutations in the alternatively spliced exons of USH1C cause non-syndromic recessive deafness	[46]
*ROM1*	*peripherin 2 like*	It is a photoreceptor specific integral membrane protein with ~35% sequence identity to *PRPH2/RDS*	[47]
*DPF3*	*double PHD fingers 3*	Its overexpression in renal cell lines increases the growth rates and alters chromatin accessibility and gene expression, leading to the inhibition of apoptosis and the activation of oncogenic pathways	[48]
*RGS6*	*regulator of G protein signaling 6*	Regulator of *G-protein signaling 6 (RGS6)* is linked to autism spectrum disorder, bipolar disorder, major depression, and schizophrenia	[49]
*SIPA1L1*	*signal induced proliferation associated 1 like 1*	*CircSIPA1L1* upregulates *ALPL* through targeting *miR-204-5p* and promotes the osteogenic differentiation of SCAPs	[50]
*PCNX1*	*pecanex homolog 1*	Plays an important role in spermatogenesis	[51]
*FASTKD2*	*FAST kinase domains 2*	Metabolic phenotyping of *FASTKD2*-deficient cells reveals impaired cellular respiration with reduced activities of all respiratory complexes	[52]
*SEMA5B*	*Semaphorin 5B*	Promotes in vivo tumor growth	[53]
*CC2D1B*	*coiled-coil and C2 domain containing 1A*	Contributes to the regulation of developmental myelination in the central nervous system	[54]
*PDE8A*	*phosphodiesterase 8A*	May indicate a role for PDE8A in cAMP signaling related to motor function	[55]
*IGF1R*	*insulin like growth factor 1 receptor*	Important effect on growth, carcass. and meat quality traits in many species	[56]
*TENM2*	*teneurin transmembrane protein 2*	Its deficiency in human adipocyte precursors leads to the induction of brown adipocyte marker genes upon adipogenic differentiation	[57]
*RRN3*	*RRN3 homolog, RNA polymerase I transcription factor*	Plays a major role in the transcriptional regulation of ribosomal DNA and cell growth	[58]
*MAD1L1*	*MAD1 mitotic arrest deficient like 1*	Deregulation of cell proliferation in avian species	[59]
*MYH11*	*myosin, heavy chain 11, smooth muscle*	Involved in vascular contractility and several autosomal dominant mutations	[60]
*TRAFD1*	*TRAF-type zinc finger domain containing 1*	Involved in interferon (IFN)γ signaling and MHC I antigen processing/presentation	[61]
*GSG1L2*	*GSG1 like 2*	This gene would have been recruited to modulate *AMPAR* function early in vertebrate evolution	[62]
*GLP2R*	*glucagon-like peptide 2 receptor*	This gene and its ligand in vertebrates has a role in embryonic intestine development	[63]
*STXBP4*	*syntaxin binding protein 4*	Regulates APC/C-mediated p63 turnover and drives squamous cell carcinogenesis	[64]
*HELZ*	*helicase with zinc finger*	Directly interacts with *CCR4-NOT* and causes decay of the bound mRNAs	[65]
*IFT22*	*intraflagellar transport 22*	Binding of *IFT22* to the intraflagellar transport complex is essential for flagellum assembly	[66]
*SNX19*	*sorting nexin 19*	Restricts endolysosome motility through contacts with the endoplasmic reticulum	[67]
*NTM*	*neurotrimin*	Mediates estrogen-induced sympathetic pruning in some peripheral targets	[68]
*PEX11G*	*peroxisomal biogenesis factor 11 γ*	Implicated in various stages of peroxisome assembly	[69]

## Data Availability

All relevant data are included in the manuscript and its additional files. The datasets used and analyzed during the current study are available from the corresponding author on reasonable request.

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
