# Peer review of "Genomics of Dwarfism in Italian Local Chicken Breeds"

_genes, 2023, doi:10.3390/genes14030633_

Round 1

Reviewer 1 Report

1 in Table 1, add the bird age and number for each sex of every breed.

2 in figure 1, the note for the breed should be clear. Especally, the same breed in this figure should be presented in the same symble, including a,b and c. e.g. PPP was different in figure1-a.

3 This research just conducted  data analysis, but not have any experiment to verify the predicted results. 

Reviewer 2 Report

This manuscript reported the candidate genes for dwarfism in three Italian chicken breeds (MERI, MUG, and PPP) by using genome-wide association study analyses. The authors claim that MERI and MUG breeds seem to share a genetic basis of dwarfism, which differentiates them from the small PPP breed.

Suggestions

1. Please clearly write the Introduction and what you need to say in each paragraph. The description of chicken breeds should combine in a paragraph.

2. L42: Can the authors describe the word “reduced body weight”?

3. This makes me confused. The authors mention that “heavy body weight and rapid growth have always been important traits within the poultry industry” but need to maintain the dwarf phenotype. Did the author mean that the dwarf phenotype will lose because of crossbreeding to increase body size?

4. L113-114: What is the PPR, PRL? The authors did not mention the full name of abbreviations before.

5. The caption should be stated below in each figure.

6. Figure 1: Some abbreviations in the caption have not followed the Figure, such as Cornuta di Caltanissetta (COR) was shown in the caption but in the figure is CORN. Please re-check. Moreover, the size of Figure1a-c is too small and difficult to see.

7. The discussion in paragraph 1 should interpret the results for readers and provide the significance of the findings. Inferring to table and figure seems to repeat the Result.

Reviewer 3 Report

Dwarf of chicken breeds is an interesting phenotype, meanwhile, parts of dwarf hens feathered with some special functions, such as food saving. The experiment explored sequence variations associated with dwarf phenotype in three Italian local chicken breeds, and some target genes were detected, which provided new sight on dwarf chicken. The manuscript was well written and data were reasonable analyzed and a certain depth of excavation. However, some proposal below may be considered.

1. Line 127-129, some phenotypic data of control population were measured in previous study, I think the data also should be listed in Table 1, which would provided more basic information and made the difference more intuitive.

2. Line 140-142, the genome-wide significance threshold was determined by the Bonferroni method considering the number of tests, and there exist 2 threshold lines in Manhattan plot, as figure 2 and others. Please add relevant description.

3. The plots of PCA and MDS in Figure 1 are of in low dpi, which make it difficult to view information of each breed, please increase the resolution of the picture.

4. Line 20-22, in abstract, 541 chicken from 23 local breeds were mentioned, in fact, only 7 breeds were used for analysis in the current study, and I think the description is a little misleading. And in Figure 1, how the 7 breeds were selected form the 23 breeds? Would Figure 1. a) was redundant and should be removed ?

5. In the study, the dwarf phenotype of the 3 breeds are more likely to be qualitative or quantitative traits? Some introduction or discussion content were expected to added.

6. The data Manhattan plot of Figure 2 and Figure 3 points are somewhat scattered, which may be caused by relatively small sample size, therefor resulting in certain false positive data. So validation samples of target SNPs added in the current experiment would be appreciated and convincing.

7. Line 51-53, the reference for creeper dwarfism and IHH was not original article or adequate, please add reference of Jin et al. (Deletion of Indian hedgehog gene causes dominant semi-lethal Creeper trait in chicken, 2016) or others. 

Round 2

Reviewer 1 Report

figure1-a,b and the notes were still not clear.

Author Response

Figure 1 and the notes were revised and improved.

Figure 1b has changed in Figure S1.

The citations in the manuscript were revised accordingly.

Reviewer 3 Report

I think the manuscript has been revised as suggested, which may make it more comprehensive.

Author Response

Many thanks for your comment.